Induced sensitivity of Bacillus subtilis colony morphology to mechanical media compression

Polka Jessica K.
Silver Pamela A. pamela_silver@hms.harvard.edu
Systems Biology Department, Harvard Medical School , USA
Wyss Institute for Biologically Inspired Engineering, Harvard University , USA
Chen Guo-Qiang
Electronic publication date: 2014 Sep 30
Publication date: 2014
Volume: 2
Electronic Location ID: e597
Received 2014 Aug 9; Accepted 2014 Sep 6
Copyright: © 2014 Polka and Silver
Copyright year: 2014
Copyright holder: Polka and Silver
License: This is an open access article distributed under the terms of the Creative Commons Attribution License, which permits unrestricted use, distribution, reproduction and adaptation in any medium and for any purpose provided that it is properly attributed. For attribution, the original author(s), title, publication source (PeerJ) and either DOI or URL of the article must be cited.
License URL: https://creativecommons.org/licenses/by/4.0/

Keywords: Mechanotaxis, Elasticotaxis, Bacillus mycoides, Bacillus subtilis, Chain length, EDTA, Swarming

Funding: Office of Naval Research MURI N00014-11-1-0725 JKP was supported by a fellowship from the Jane Coffin Childs Fund. The work was also supported by an Office of Naval Research MURI Grant N00014-11-1-0725. The funders had no role in study design, data collection and analysis, decision to publish, or preparation of the manuscript.

==============================
Bacteria from several taxa, including Kurthia zopfii, Myxococcus xanthus, and Bacillus mycoides, have been reported to align growth of their colonies to small features on the surface of solid media, including anisotropies created by compression. While the function of this phenomenon is unclear, it may help organisms navigate on solid phases, such as soil. The origin of this behavior is also unknown: it may be biological (that is, dependent on components that sense the environment and regulate growth accordingly) or merely physical.

Here we show that B. subtilis, an organism that typically does not respond to media compression, can be induced to do so with two simple and synergistic perturbations: a mutation that maintains cells in the swarming (chained) state, and the addition of EDTA to the growth media, which further increases chain length. EDTA apparently increases chain length by inducing defects in cell separation, as the treatment has only marginal effects on the length of individual cells.

These results lead us to three conclusions. First, the wealth of genetic tools available to B. subtilis will provide a new, tractable chassis for engineering compression sensitive organisms. Second, the sensitivity of colony morphology to media compression in Bacillus can be modulated by altering a simple physical property of rod-shaped cells. And third, colony morphology under compression holds promise as a rapid, simple, and low-cost way to screen for changes in the length of rod-shaped cells or chains thereof.

Introduction

Response of bacterial colony morphology (i.e., orientation of growth) to small mechanical perturbations of growth media was first noted in Kurthia, a gram-positive genus notable for its striking feather-like morphology on gelatin slant cultures (Sergent, 1906; Sergent, 1907; Jacobsen, 1907; Stackebrandt, Keddie & Jones, 2006). A similar compression response has been reported in Myxococcus xanthus, where the phenomenon is dependent on adventurous motility, a flagellum- and pili-independent movement system (Stanier, 1942; Fontes & Kaiser, 1999; Nan et al., 2014). Recently, the soil bacterium Bacillus mycoides was also shown to be sensitive to media perturbations (Stratford, Woodley & Park, 2013). Interestingly, this compression response seems to occur by two different mechanisms: whereas individual Myxococcus xanthus dynamically reorients individual cells along lines of compression (Dworkin, 1983), Bacillus mycoides instead gradually reorients the tips of chained cells as it grows (Stratford, Woodley & Park, 2013).

The function of compression response is not known, but it has been suggested to aid navigation in natural environments on solid phases, like soil (Dworkin, 1983). It has also been proposed as a potential tool for engineering applications in sensing environmental forces or generating patterns for nanofabrication (Stratford, Woodley & Park, 2013).

Here we investigate whether increasing the length of chains of cells can induce compression sensitivity in an otherwise compression-insensitive species, B. subtilis. We employ a mutant of B. subtilis that forms long chains of cells (much like B. mycoides) and also deplete divalent cations in the media with EDTA; Mg2+ is thought to be important for cell wall integrity. B. subtilis deprived of magnesium accumulates cell wall precursors, (Garrett, 1969), and magnesium is known to bind to components of the cell wall (Heckels, Lambert & Baddiley, 1977). Notably, high magnesium concentrations can restore rod shape to cells with mutations in MreB, MreD, and PonA—all genes involved in cell wall synthesis (Rogers, Thurman & Buxton, 1976; Rogers & Thurman, 1978; Murray, Popham & Setlow, 1998; Formstone & Errington, 2005).

Materials and Methods

Time lapse microscopy

2% LB agar was cut into approximately 10 mm × 10 mm squares and inoculated with 1 µl of liquid culture. The pad was then wedged, in a glass-bottomed dish (P35G-1.5-20-C; MatTek Corp.), between two plastic coverslips (Rinzl Plastic Coverslips, Size 22 × 22 mm; Electron Microscopy Science) manually bent in half at a 90° angle. Thus, half of each plastic coverslip made contact with the bottom of the dish, while the other half made contact with the agar pad. After placing a drop of approximately 50 µl of water on top of each plastic coverslip to maintain humidity in the dish, the MatTek dish was sealed with parafilm (this setup is illustrated in Fig. 1A). Cells (Table 1) were grown for approximately 6 h at room temperature (approximately 23°) during a timelapse acquisition on a Nikon TE 2000 microscope equipped with an Orca ER camera, a 20 × phase contrast objective, and Perfect Focus. A large area of the sample was composited with automatic image stitching by Nikon Elements AR. Areas toward the center of the pad were selected for imaging.

Figure 1 Microscopic morphology of B. mycoides and B. subtilis under compression.

(A) Cells from liquid culture were applied to the bottom of an agarose pad compressed between plastic coverslips in a MatTek dish. Black arrows indicate direction of compression throughout. (B) Striations visible in agar surfaces. (C) Montages of timelapses of B. mycoides, B. subtilis PY79, and B. subtilis σD::tet. Note the striations visible in the agarose running perpendicular to the direction of compression.

Table 1 Strains used in this study.

Designation	Description	Reference	
B. subtilis PY79	Lab strain	Bacillus Genetic Stock Center 1A747	
B. subtilis σD::tet	RL4169, DS323	Kearns & Losick (2005)	
B. mycoides		ATCC 6462	

Plate compression

Microtiter format plates were prepared with LB + 2% agar. 24 h after plates were poured, sterilized polystyrene spacers (each 0.080″ thick, for a total compression of 0.16″ or 4.1 mm, equivalent to 4.8% compression) were inserted along the long dimension. Plates were stored at 37° for 24 h, then inoculated from colonies grown on LB agar. Plates were incubated for 2–3 days at 30°, as the time required to reach colony dimensions >8 mm varied with EDTA concentration. After incubation, plates were imaged with a gel imager and colony dimensions measured with FIJI (Schindelin et al., 2012).

Cellular morphology

Colonies were grown on LB + 2% agar containing either 0 or 125 µM EDTA. After 24 h of incubation at 30°, cells from the edges of colonies were transferred directly to LB + 2% agar pads for imaging with the rounded bottoms of 0.6 µl centrifuge tubes. To each pad, 1 µl of an aqueous solution containing 10 µg/ml FM4-64 (Invitrogen) was added. Cells were imaged with a 100 × phase contrast objective, and cell and chain lengths were measured manually with spline-fitted segmented lines in FIJI. Two-sample KS tests were performed (Kirkman, 1996).

Results

We first noted weak compression response of B. subtilis under the microscope. Unlike B. mycoides, B. subtilis colonies remain circular under compression under normal conditions. However, our microscopy assay (Fig. 1A) revealed that at small length scales (<100 µm), B. subtilis cells display short-range alignment perpendicular to the direction of compression (marked with black arrows in Figs. 1A–1C). Noting that the alignment is disrupted over longer length scales, we sought conditions under which B. subtilis cells might behave more similarly to B. mycoides. We noted that the chains of B. subtilis PY79 appeared shorter than those of B. mycoides, with the former reaching a maximum of approximately 300 µm (Fig. 1C), while the latter can extend for millimeters (Stratford, Woodley & Park, 2013).

To increase chain length, we used B. subtilis σD::tet, a mutant that does not switch from sessile to motile states, and thus grows in long chains of cells (Kearns & Losick, 2005). To further perturb cell separation, we added EDTA to the growth medium.

To study colony morphology of B. subtilis under compression at the macroscopic scale with reproducible compression conditions, we prepared microtiter plates with LB + 2% agar and wedged polystyrene spacers between the agar and an edge of the plates (Fig. 2A). We inoculated the agar with colonies of B. mycoides, B. subtilis PY79, and B. subtilis σD::tet. Under 4.8% compression, B. mycoides forms elongated colonies as reported, (Stratford, Woodley & Park, 2013) while, without EDTA, B. subtilis colonies are round (Fig. 2A). With the addition of EDTA to the media, both B. subtilis PY79 and σD::tet display a compression response (Fig. 2B). This is dependent on the degree of compression; at 2.4% compression, both B. subtilis strains formed round colonies (data not shown).

Figure 2 B. mycoides and B. subtilis colony morphology under compression.

(A) A microtiter plate inoculated with B. mycoides and B. subtilis. The two white bars at the top of the image of the plate are polystyrene spacers, totaling 4.8% of the plate height. Black arrows indicate direction of compression throughout. (B) Representative images of B. subtilis PY79 and σD::tet colonies grown on compressed agar with varying EDTA concentrations. Scale bar, 1 cm. (C) Plot of colony shape ratio (i.e., colony measurement perpendicular to the dimension of compression/colony measurement parallel to the dimension of compression) as it varies with EDTA concentration. (D) Same as in C but with axes scaled to emphasize relative effect of PY79 and σD::tet, individual data points removed, and 95% CI error bars added. The σD::tet data has been shifted by 2 x-axis units to better display the error bars. For each condition, n > 11. Source data for this figure can be found in Data S1.

We next quantified this effect over several colonies under each EDTA condition at 4.8% compression. Bacillus mycoides forms colonies 4–4.5 × larger in the dimension perpendicular to the direction of compression than parallel to it regardless of EDTA concentration (Fig. 2C). In comparison, the effect in B. subtilis is relatively small, and this effect scaled with EDTA concentration (Fig. 2C). The EDTA effect was stronger for the σD::tet strain; at 125 µM EDTA, compressed σD::tet colonies were an average of 1.64 × larger in the direction of compression (n = 17, standard deviation 0.21), while PY79 colonies were an average of 1.23 × larger (n = 16, standard deviation 0.20). While the difference in colony size ratio between 0 and 125 µM EDTA for PY79 is significant by a two-tail t-test (p < 0.02), the difference between these concentrations for σD::tet is highly significant (p < 0.00001).

Furthermore, colonies from all three strains, but especially B. subtilis PY79 and B. mycoides, grow at slower rates with increased EDTA concentration. The difference in growth rate on EDTA may be attributable either to species- and strain-specific sensitivity to EDTA, or (in the case of PY79 and σD::tet) to differences in sensitivity between swimming and swarming cells.

To understand how EDTA could affect compression response, we imaged cells taken directly from the edges of colonies on solid media containing either 0 µM (Figs. 3A–3C) or 125 µM EDTA (Figs. 3D–3F). The chains of B. subtilis cells, both PY79 and σD::tet, are longer on 125 µM EDTA, but cell lengths, as delineated by the membrane dye FM4-64, are only marginally different. Quantification of ∼300 chain and cell lengths for each strain under each condition (Fig. 4) reveals that B. subtilis chain lengths increase dramatically with the presence of EDTA, while B. mycoides chain lengths decrease slightly, suggesting that the EDTA effect on cell separation is specific to B. subtilis (Table 2).

Figure 3 Cellular morphology with and without EDTA.

(A)–(C) B. mycoides, B. subtilis PY79, and B. subtilis σD::tet, respectively, growing on LB agar containing 0 µM EDTA. (D)–(F) As above on 125 µM EDTA. In all images, phase contrast channel is in red, and FM4-64 is in green. Scale bar, 10 µm. Source data for this figure can be found in Data S2.

Figure 4 Quantification of chain and cell lengths with and without EDTA.

(A) Cell lengths of B. mycoides on 0 µM (hollow bars) and 125 µM EDTA (grey bars). (B) Chain lengths of B. mycoides. (C) Cell lengths of B. subtilis PY79. (D) Chain lengths of B. subtilis PY79. (E) Cell lengths of B. subtilis σD::tet. (F) Chain lengths of B. subtilis σD::tet. Source data for this figure can be found in Data S2.

Table 2 Properties of cell and chain length measurement distributions.

	Cell length	Chain length	
	0 µM EDTA mean (µm)	125 µM EDTA mean (µm)	KS test maximum difference	0 µM EDTA mean (µm)	125 µM EDTA mean (µm)	KS test maximum difference	
B. mycoides	4.01 (st dev 1.54)	4.33 (st dev 2.04)	D = 0.1044, P = 0.051	9.19 (st dev 4.81)	6.60 (st dev 3.09)	D = 0.2959, P < 0.001	
B. subtilis PY79	3.18 (st dev 1.03)	4.18 (st dev 1.93)	D = 0.2866, P < 0.001	3.94 (st dev 1.38)	13.71 (st dev 7.23)	D = 0.8505, P < 0.001	
B. subtilis σD::tet	4.23 (st dev 3.20)	4.12 (st dev 2.18)	D = 0.2413, P < 0.001	7.50 (st dev 3.36)	21.99 (st dev 18.1)	D = 0.5633, P < 0.001	

Discussion

These results suggest that the phenomenon of colony orientation under compression can be induced in the model organism B. subtilis. In contrast to Bacillus mycoides (the transformation of which has been reported only anecdotally in the literature (Di Franco et al., 2002)), the genetic tractability of B. subtilis will facilitate engineering of compression sensitive bacteria for use as environmental sensors or guides for nanofabrication (Stratford, Woodley & Park, 2013).

Furthermore, the fact that that colony orientation on compressed media is generalizable indicates that it is likely to be a physical phenomenon. While we cannot exclude the involvement of biological components, any such components are certainly not exclusive to B. mycoides. Furthermore, the A-motility required for compression response in myxobacteria is not a requirement for all types of compression response (Nan et al., 2014). Instead, it is likely that this compression response requires physical factors like rod length, surface friction, cell stiffness, and tip vs. isotropic growth pattern.

Long rod length is a common feature of two prototypical compression responders, Bacillus mycoides and Kurthia sp., which both grow as long chains of cells (Di Franco et al., 2002; Stackebrandt, Keddie & Jones, 2006). As seen in microscopy of B. mycoides, the absence of cell separation allows the bacteria to find and maintain a direction of compression. This same chaining property is responsible for the baroque colony morphology of B. mycoides: mutants that do not display this colony morphology have shorter chain lengths (Di Franco et al., 2002). Thus, compression response may be driven by the same mechanisms that influence colony morphology under normal conditions; these mechanisms influence the manner in which cells explore and colonize their environment, and may be of critical importance in soil environments.

In the case of B. subtilis, the increase in compression sensitivity is based on chain length (as a σD mutant responds more than PY79, and both respond more strongly in the presence of EDTA, which also increases chain length). Though EDTA likely affects multiple cellular processes, the role of Mg2+ in cell wall formation is clear (Formstone & Errington, 2005). In particular, peptidoglycan hydrolases called autolysins are implicated in separation of cells after septation. Some of these autolysins, such as LytC, D, and F, are under the control of σD (Chen et al., 2009). However, LytC expression can also be driven by σA (Lazarevic et al., 1992), and this 50 kDa amidase is activated by addition of Mg2+in vitro (Foster, 1992). We speculate that this magnesium dependence of LytC and its regulation by a second sigma factor may explain why EDTA treatment further increases chain length in σD::tet cells. In addition to LytC, EDTA may be acting on other autolysins not regulated by σD (such as LytE or YwbG) (Smith, Blackman & Foster, 2000). The insensitivity of B. mycoides chain length to EDTA (Fig. 4 and Table 2) may be explained by species-specific differences in autolysins.

Inhibition of cell separation may not be the only relevant effect of EDTA, however. For example, perhaps depletion of Mg2+ changes the rigidity of cells such that they more readily align with the isotropic agar surface (Fig. 1B). An exhaustive understanding of EDTA’s effects on the mechanical properties of B. subtilis walls, as well as a mechanistic understanding of how it increases chain length, remains to be attained.

The relatively weak maximal compression response we achieved with B. subtilis compared to B. mycoides suggests that factors other than chain formation limit the compression response of B. subtilis. Indeed, filament or chain formation alone must not be sufficient for compression response, as some fungi and actinomycetes grow with this morphology but do not display the response (Stratford, Woodley & Park, 2013). We suggest that friction with the agar surface may play a significant role. In micrographs of B. subtilis under compression, the chains of cells appear more buckled than those of B. mycoides (Fig. 1C); perhaps friction prevents the distal ends of the chain from sliding along to accommodate new growth from the middle of the chain. This buckling disrupts adjacent chains and is likely to lead to a more disorganized colony morphology. By contrast, B. mycoides chains elongate at a rate of 0.5 mm per hour, suggesting that the cells at the tip of the chain are being pushed forward by growth from the middle of the chain (Stratford, Woodley & Park, 2013). In the future, further modifications, perhaps increasing surfactin production, may increase the magnitude of this response in B. subtilis. Additionally, we note that another contributing factor may be the growth pattern of this organism. Whereas B. mycoides elongates from its tips (Turchi et al., 2012), B. subtilis inserts cell wall isotropically along its length (Tiyanont et al., 2006).

Finally, because B. subtilis compression response depends on chain length, we propose that under some circumstances, colony morphology under compression could serve as a simple, high-throughput assay for perturbations to bacterial cell length and chain formation.

Supplemental Information

Data S1 Calculated colony size ratios on compressed agar

These measurements were made with FIJI (see “Materials and Methods”) from a dataset available at http://dx.doi.org/10.6084/m9.figshare.1133794. This is the source data for Figs. 2C–2D.

Click here for additional data file.

Data S2 Cell and chain length measurements and histograms

These measurements were made with FIJI (see “Materials and Methods”) from a dataset available at http://dx.doi.org/10.6084/m9.figshare.1133867. This is the source data for Fig. 4.

Click here for additional data file.

We thank Ethan Garner (Harvard University), Michael Baym (Harvard Medical School) and Ariel Amir (Harvard University) for helpful discussions. We are grateful to Stephanie Hays and Anna H. Chen (Harvard Medical School) for critical reading of the manuscript, and to James P. Stratford (Nottingham) and two anonymous reviewers for their insightful comments.

Additional Information and Declarations

Competing Interests

Author Contributions

Data Deposition

The authors declare there are no competing interests.

Jessica K. Polka conceived and designed the experiments, performed the experiments, analyzed the data, contributed reagents/materials/analysis tools, wrote the paper, prepared figures and/or tables, reviewed drafts of the paper.

Pamela A. Silver wrote the paper, reviewed drafts of the paper.

The following information was supplied regarding the deposition of related data:

Polka, Jessica (2014): Bacillus mycoides and subtilis colony mophology on compressed agar. figshare.

http://dx.doi.org/10.6084/m9.figshare.1133794

Retrieved 18:47, Aug 09, 2014 (GMT)

Polka, Jessica (2014): Bacillus mycoides and subtilis cell morphology. figshare.

http://dx.doi.org/10.6084/m9.figshare.1133867

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
