# Peer review of "Induced sensitivity of Bacillus subtilis colony morphology to mechanical media compression"

_PeerJ, doi:10.7717/peerj.597_

## Round 0.1 · original submission · Minor Revisions

Dear Pam,

Thank you for submitting your paper to PeerJ.

Your paper has been reviewed by three experts in the field, Reviewers 1 and 3 are positive on your paper, reviewer 2 has some reservations, please address these concerns and submit a revised version of this paper as soon as you could.

Best regards

George
George Guo-Qiang CHEN (Ph.D.)
Professor of Microbiology and Biomaterials
Department of Biological Sciences and Biotechnology
School of Life Sciences
Tsinghua University
Beijing 100084 China
Tel: +86-10-62783844
Fax: +86-10-62794217
e-mail: chengq@mail.tsinghua.edu.cn

·

Basic reporting

The study demonstrates a close association between cell chain formation and elasticotaxis. This is shown using the response of Bacillus subtilis colony morphology to alterations in cell chain length achieved by using a mutant and a chemical modification (EDTA) of a solid growth medium. Bacillus subtilis has not previously been observed to respond to mechanical force and a clear shift to elasticotaxis at the colony level is demonstrated as a result of both modifications.
The author’s main conclusions are that chain formation is required for elasticotaxis while they also conclude that as a result elaticotaxis is likely to be a purely physical process.

The study is well written and clearly presented.

The work uses an elegant and innovative experimental design.

The description given of the materials and methods used is clear.

The results are novel and constitute an advance in understanding in an area where
there has been little progress made in providing mechanistic explanations.

The manuscript is technically sound and certainly deserving of publication.

Experimental design

The experimental design is novel, innovative and meets all of Peerj's criteria for publication.

Validity of the findings

As a conclusion of their work the authors suggest elasticotaxis is a purely physical process.
This conclusion oversimplifies what may in fact be a complex process and may deter rather than encourage further scholarly investigation. It would be better to state precisely what has been shown i.e. that chain formation appears to be a prerequisite for elasticotaxis in bacteria. Further elaboration of this position is given below:

B. mycoides and B. subtilis are related organisms.

Can the authors be sure no sensing systems are involved?

Fungi and Actinomycetes form filaments yet have not been observed to exhibit elaticotaxis in the literature or during the course of several experiments conducted during the preliminary work for Stratford et al (2014).

It is not known that wild-type B. subtilis lacks the intrinsic capacity to sense force, only that it does not normally display morphological signs of the response at a colony level during culture on agar.

The authors also state that they were prompted to investigate the effect of cell chain length on elasticotaxis by an initial observation of the phenomenon, albeit weak, in wild type B. subtilis. This suggests that B. subtilis is intrinsically elasticotactic but that the response is latent during development of colony morphology under normal culture conditions.

Bacillus mycoides colony shape ratio is around 5 while increasing the length of B. subtilis cell chains results in a ratio of around 1.6. This is a dramatic variation associated with a species difference.

Assuming chain length is ultimately responsible for determining the magnitude of the elasticotactic response then mechanisms for preventing buckling are important biological factors e.g. lubrication? More substantial cell wall structure?

This paper discovers elasticotaxis in B. subtilis and demonstrates that the magnitude of its elastotactic response is related to cell chain length.

The manuscript would be improved by clarifying some of the language used to reflect what has been definitively shown.

Additional comments

Some helpful suggestions and future directions:

Does B. subtilis posses the type of internal propulsion associated with myxobacterial A-motility (Nan et al, 2011)? Is it therefore possible to definitively exclude this as a requirement for elasticotaxis?

Do very distantly related rod-shaped bacteria also exhibit elasticotaxis when induced to form chains?

It may be worth emphasizing the probable gliding of long chains under the propulsive force of their own expansion. The speed of tip growth, 0.5mm/hour, recorded in Stratford et al (2014) suggests B. mycoides chains are in motion rather than just growing at the tips. In addition to filament formation this growth driven gliding may be a prerequisite for elasticotaxis. It would explain why actinomycetes and fungi do not exhibit the phenomenon as their filaments undergo predominantly apical growth rather than expansion distributed along the length of the chain (Goriely and Tabor, 2008).

Another chain forming organism exhibiting a weakly elasticotactic response is Clostridium sporogenes, this observation was not included in the original mycoides manuscript as the response was weak, only visible at colony peripheries and B.mycoides produced much clearer results. This personal observation supports the conclusion that chain formation is likely to be a necessary prerequisite for elasticotaxis. This comment is not meant to contribute to the paper but suggests another model organism for further work in this area.

Signed

James P Stratford


References

Goriely A, Tabor M (2008) Mathematical modelling of hyphal tip growth. Fungal Biology Reviews 22(2): 77–83. doi: 10.1016/j.fbr.2008.05.001

Nan B, Chen J, Neu JC, Berry RM, Oster G, et al. (2011) Myxobacteria gliding motility requires cytoskeleton rotation powered by proton motive force. Proc Natl Acad Sci USA 108: 2498–2503. doi: 10.1073/pnas.1018556108

Stratford JP, Woodley MA, Park S. 2013. Variation in the Morphology of Bacillus mycoides Due to Applied Force and Substrate Structure. PLoS ONE 8:e81549.

Reviewer 2 ·

Basic reporting

Media compression may induce some kinds of swarming bacteria to align their growing cells in response to the change of media. This paper described an interesting observation that B. subtilis, a bacterium weakly responds to media compression, can be induced to grow in response to media compression, if using a mutant plus the addition of EDTA to the media. the results cue some important conclusions and uses.

Experimental design

Preliminary, and needs more experiments.

Validity of the findings

Not consolidated.

Additional comments

In general, this paper is preliminary, and need more experimental supports for conclusions, which, at present, are rather exaggerated and only suggestions. For example, the first of the three conclusions, i.e. the wealth of genetic tools available to B. subtilis will provide a genetically tractable chassis for engineering compression sensitive organisms in the future, has no direct relationship to the finding. The second, i.e. the sensitivity of colony morphology to media compression in Bacillus is a physical rather than biological phenomenon dependent on a simple physical property of rod-shaped cells, is just a suggestion, which needs more consolidated supporting experiments, e.g. using some other chelants to define functions of magnesium or others; if magnesium functioned, then how about the growth of the σD::tet mutant on magnesium-free media?

Reviewer 3 ·

Basic reporting

No Comments

Experimental design

No Comments

Validity of the findings

No Comments

Additional comments

The manuscript describes colony morphology under compression could serve as a simple, high-throughput assay for perturbations to bacterial cell length and chain formation under some circumstances. The increase in compression sensitivity is based on chain length in B. subtilis. And the respond more strongly in the presence of EDTA. There are several concerns that the authors should take into account in the revision.

1. Figure 2B, With the addition of EDTA to the media, both B. subtilis PY79 and σD::tet display a compression response. How about the colonies shape of the Bacillus mycoides with the addition of different concentration EDTA to the media? Since it is likely to be a physical phenomenon, why B. subtilis PY79 colony morphology becomes gradually smaller with increasing EDTA concentration, and B. subtilis σD::tet were not?

2. Figure 2C, the effect in B. subtilisis relatively small. Bacillus subtilis colonies were a maximum of approximately 1.5x larger in the direction perpendicular to compression. This 1.5x larger Bacillus subtilis colonies are PY 79 or σD::tet? How about the standard deviation? 1.64x larger Bacillus subtilis colonies were achieved after add of 125uM EDTA, is the difference significant?

---

## Round 0.2 · accepted · Accept

Dear Pam,

I am pleased to inform you that your revised paper has been accepted by PeerJ.

Best regards

George

George Guo-Qiang CHEN (Ph.D.)
Academic Editor, PeerJ.
Professor of Microbiology and Biomaterials
Department of Biological Sciences and Biotechnology
School of Life Sciences
Tsinghua University
Beijing 100084 China
Tel: +86-10-62783844
Fax: +86-10-62794217
e-mail: chengq@mail.tsinghua.edu.cn